# Metabolites and Bioactivity of the Marine *Xestospongia* Sponges (Porifera, Demospongiae, Haplosclerida) of Southeast Asian Waters

**DOI:** 10.3390/biom13030484

**Published:** 2023-03-06

**Authors:** Fikri Akmal Khodzori, Nurzafirah Binti Mazlan, Wei Sheng Chong, Kuan Hung Ong, Kishneth Palaniveloo, Muhammad Dawood Shah

**Affiliations:** 1Borneo Marine Research Institute, Universiti Malaysia Sabah, Jalan UMS, Kota Kinabalu 88450, Malaysia; 2Institute of Ocean and Earth Sciences, Advanced Studies Complex, Universiti Malaya, Kuala Lumpur 50603, Malaysia; 3Centre for Natural Products Research and Drug Discovery (CENAR), Level 3, Research Management & Innovation Complex, Universiti Malaya, Kuala Lumpur 50603, Malaysia

**Keywords:** Sponge, Demospongiae, *Xestospongia*, metabolites, bioactivity, Southeast Asian waters

## Abstract

Sponges are aquatic, spineless organisms that belong to the phylum Porifera. They come in three primary classes: Hexactinellidae, Demospongiae, and Calcarea. The Demospongiae class is the most dominant, making up over 90% of sponge species. One of the most widely studied genera within the Demospongiae class is *Xestospongia*, which is found across Southeast Asian waters. This genus is of particular interest due to the production of numerous primary and secondary metabolites with a wide range of biological potentials. In the current review, the antioxidant, anticancer, anti-inflammatory, antibacterial, antiviral, antiparasitic, and cytotoxic properties of metabolites from several varieties of Southeast Asian *Xestospongia* spp. were discussed. A total of 40 metabolites of various natures, including alkaloids, fatty acids, steroids, and quinones, were highlighted in *X. bergquistia*, *X. testudinaria*, *X. muta*, *X*. *exigua, X. ashmorica and X. vansoesti*. The review aimed to display the bioactivity of *Xestospongia* metabolites and their potential for use in the pharmaceutical sector. Further research is needed to fully understand their bioactivities.

## 1. Introduction

Sponges, aquatic animals of the phylum Porifera, have existed for millions of years as the simplest multicellular organisms. They are filter feeders and are known for their unique species diversity and morphological complexity [1,2]. Sponge species number over 8000 and are found in temperate, tropical, and polar regions, inhabiting a wide range of freshwater and marine habitats [3]. They are an important source of metabolites. More than 5300 distinct metabolites produced by sponges and the accompanying microbes are known and more than 200 novel sponge metabolites are reported each year [3]. Alkaloids, fatty acids, sterols, terpenoids, polyketones, macrolides, quinines, glucosides, and peptides are a few examples of novel metabolites that have been identified from marine sponges [4,5,6,7,8,9].

There are three primary classes of sponges: Hexactinellidae, Demospongiae, and Calcarea. The Demospongiae class is the most dominant, comprising over 90% of sponge species. The genus *Xestospongia* (Petrosiidae) is widely studied due to its various primary and secondary metabolites with various biological potentials. They are also known as “giant barrel sponges” and have a large, erect, barrel-shaped appearance with variations in height, diameter, and surface complexity among distinct species [10]. However, they can be distinguished by their unique morphological characteristics, which include a thick cortex, a large central osculum, and a porous spongin skeleton [11,12,13]. Their external morphology also varied from smooth to highly digitated or lamellate surfaces (Figure 1) [10,11,14,15].

*Xestospongia* spp. are classified as follows within the phylum Porifera. Kingdom: Animalia (animals), Phylum: Porifera (sponges), Class: Demospongiae (demosponges), Order: Haplosclerida, Family: Xestospongiidae, Genus: *Xestospongia.* There are over 30 *Xestospongia* spp., including *X. bergquistia* (Fromont, 1991) [16], *X. testudinaria* (Lamarck, 1815) [17], *X. muta* (Schmidt, 1870) [18], *X. exigua* (Kirkpatrick, 1900) [19], *X. ashmorica* (Hooper, 1984) [20] and *X. vansoesti* [21] etc.

*Xestospongia* spp. can be found in a variety of habitats in the Southeast Asia region. They are typically found in shallow tropical coral reefs but can also be found in deeper waters. They are often found attached to sand, rocks, corals, or other benthic substrates [22,23,24,25,26]. In terms of ecology, *Xestospongia* spp. play a key role in the coral reef ecosystem. They are filter feeders, which means they filter water through their bodies to obtain food particles. The huge size of these sponges is particularly crucial because body size is mechanistically connected to pumping and nutrition cycling. This process also helps to remove excess nutrients and sediment from the water, which can help maintain the overall health condition of the coral reef ecosystem [27,28]. *Xestospongia* spp. also provide important habitats for other organisms, such as algae, bacteria, fish, crustaceans, and other invertebrates, which use the sponges as a place to hide, breed, and forage for food (Figure 2). Some *Xestospongia* spp. also form symbiotic relationships with other prokaryotes [23,29,30,31].

However, *Xestospongia* spp. are facing potential threats to their survival. These include pollution, rise in temperature due to climate changes and associated diseases such as generalised necrosis, cyclic spotted bleaching, sponge orange band, tissue hardening condition and tissue wasting disease [32,33,34,35]. These threats can have negative impacts on the overall health of coral reef ecosystems and therefore on *Xestospongia* spp. as well [36,37].

*Xestospongia* species are found in tropical and subtropical waters throughout Southeast Asia (Figure 3). They are noticed in several places in Indonesia, including Pecaron Bay, Pasir Putih, Situbondo, East Java, Sabang Island, North Sulawesi, and Bandung. They can frequently be seen in deep waters, lagoons, and coral reefs [38,39,40,41]. *Xestospongia* spp. are observed in the Philippines in several places, including the Manila Channel off Mindoro Island [42]. They are located in Malaysia in several places, including Sepanggar and Gaya Islands, Sabah, Mentigi Island, Johor, Bidong Island, Terengganu, and Langkawi [43,44,45]. In Thailand, they are found in Chon Buri and Rayong Provinces [46] while in Vietnam, they are found in Ha Long Bay and Khanh Hoa [47,48].

It is also worth noting that the diversity of *Xestospongia* spp. in Southeast Asia is not well understood, and more research is needed to fully understand the diversity, distribution, and abundance of these species in the region.

This review comprehensively addresses the secondary metabolites of marine sponges of the genus *Xestospongia* in Southeast Asian waters with potential bioactive properties, including antioxidant, anticancer, antiplasmodial, antibiotic, antibacterial, antifungal, cardiotonic, cytotoxic, antimalarial, and antihelminthic properties. The review collects and compares information from peer-reviewed articles from 1974 to 2022 on secondary metabolites isolated from *Xestospongia* spp. The information was retrieved from several internet databases (PubMed, Web of Science, Scopus, and registries including dissertations and proceedings). The database was searched for marine sponges, *Xestospongia*, Indo-Pacific, Southeast Asia, the South China Sea, bioactivity, morphotypes, haplotypes, secondary metabolites and marine natural products.

## 2. Secondary Metabolites in *Xestospongia* spp. and Their Bioactivity

In Southeast Asian waters, *Xestospongia* spp. has been found to produce a variety of secondary metabolites of various nature including alkaloids, steroids, fatty acids, quinone etc. These compounds have been shown to have antioxidant, anti-inflammatory, antiparasitic, antitumor, and antimicrobial properties [4,5,6,49,50,51,52,53,54,55,56]. However, more research is needed to fully understand these metabolites’ bioactivity and potential uses. Some of the important *Xestospongia* spp. have been discussed below.

### 2.1. Xestospongia bergquistia (Fromont, 1991)

*X. bergquistia* is an abundant member of the coral reef community found in the Philippines, Indian Ocean, Indonesia, and Malaysia [15,57]. Recent studies have isolated three unique pentacyclic polyhydroxylated steroids, known as xestobergsterol A (**1**), B (**2**), and C (**3**), from the methanol/toluene extract of this species. These compounds are notable for being the first steroids to have five carbocyclic rings. Compounds **1** and **2** have been found to have anti-inflammatory activity, with both displaying potent inhibition of histamine release from rats’ mast cells induced by anti-IgE (Immunoglobulin E) in a dose-dependent manner [4,5,6]. The anti-inflammatory activity of the compound **1** was found to be approximately 5200 times stronger than that of disodium cromoglycate, a commonly used antiallergy medication. In addition to their anti-inflammatory properties, compounds **1** and **3** were also found to exhibit cytotoxic activity against L-1210 murine leukaemia cells. Specifically, they displayed IC_50_ values of 4.0 μg/mL and 4.1 μg/mL, respectively. However, compound **2** was found to have negligible cytotoxic effects [49,50]. The chemical structures of compounds **1**-**3** are shown in Figure 4.

Overall, these findings suggest that compounds **1-3** isolated from *X. bergquistia* have potential therapeutic applications in the fields of anti-inflammatory and cancer treatment. However, further research is needed to fully understand the properties and potential uses of these compounds.

### 2.2. Xestospongia muta (Schmidt, 1870)

*X. muta* is a species of sponge found on Sabang Island, Indonesia, and other parts of the world. Studies have shown that the tissues of this sponge contain symbionts of the *Synechococcus* group, which are a type of microorganism that lives in symbiotic relationships with other organisms [37,38].

A manzamine alkaloid, named manzamine C (**4**) (Figure 5), was isolated from *X. muta*, and displayed inhibition activity against human pancreatic cell carcinoma (PANC-1) under glucose starvation conditions with IC_50_ values of 10 μM, whereas no growth inhibition was observed up to 100 μM under the general culture conditions. Additionally, the compound **4** also exhibited strong antileishmanial and antimalarial activity against drug-sensitive and drug-resistant strains of *Plasmodium* [58,59]. These findings suggest that compound 4 has the potential as a treatment for these diseases, and further research is needed to explore this possibility.

In addition to the pure secondary metabolites the fractions of the solvent extract of *X. muta* also displayed protective properties. The *X. muta* collected in coastal Terengganu (Malaysia) was found to have cardiovascular protective properties. The study showed that various fractions of this species contain various fatty acids with cardiovascular protective activity, with Fraction-7 being the most notable. Fraction 7 was obtained from the methanol extract of *X. muta,* and gas chromatography and mass spectrometry (GCMS) analysis of this fraction indicated the presence of 58 compounds. In vitro research in HepG2 cells displayed that fraction-7 of *X. muta* boosted the expression of Scavenger receptor class B type I (*SR-BI*) mRNA by 129%. *SR-BI* is the primary receptor for high-density lipoprotein (HDL) cholesterol, which is crucial for preventing atherosclerosis [44].

### 2.3. Xestospongia exigua (Kirkpatrick, 1900)

*X. exigua* is abundant in tropical Southeast Asia, the Western South Pacific, Papua New Guinea, and Australia [53,60,61]. In natural product chemistry, *X. exigua* has attracted a lot of attention due to the diversity of secondary metabolites that have been isolated from it. Since the 1980s, over 24 different bioactive metabolites have been identified from this species, and these metabolites have been found to display a wide range of bioactivities, including vasodilation, cytotoxicity, and antibacterial effects [53,61,62,63,64,65].

The metabolites identified in *X. exigua* exhibited diverse chemical structures, including alkaloids, quinones, and sterols [61,62,63]. The compounds (+)-xestospongin (A–D) (**5-8**) and (+)-araguspongines K (**9**) and L (**10**) exhibited vasodilation activity [62,63]. Similarly, xestosin A (**11**), a new bis-quinolizidine alkaloid, was isolated from *X. exigua* collected in Papua New Guinea [60]. Exiguamine A. (**12**) displayed inhibition of indoleamine-2,3- dioxygenase [64]. Other compounds isolated from *X. exigua* such as motuporamines A- I (**13**-**21)** displayed cytotoxicity against human cancer cell lines [65]. According to the literature, Motuporamines G, H and I (**19**-**21**) shared identical chemical structures and varied in chemical shifts. Halenaquinone (**22**) displayed antibacterial properties [52]. Exiguaquinol (**23**) showed inhibition of *Helicobacter pylori* glutamate racemase (MurI) with an IC_50_ value of 4.4 μM. MurI catalyses the conversion of L and D-glutamate, supplying D-glutamate for integration into the elongating peptidoglycan chain that constitutes the cell walls [51]. Clionasterol (**24**) and 5α,8α-Epidioxy-24αethylcholest-6-en-βb-ol (**25**) were reported to inhibit the human complement system [53]. The compound **25** also exhibited anti-inflammatory properties. At the mRNA and protein levels, **25** reduced the liposaccharide-induced production of inflammation mediators involved in the NF-κB pathway, including tumour necrosis factor-alpha (TNF-α), interleukin 6 (IL-6), and cyclooxygenase-2 (COX-2) [66]. The chemical structures of compounds reported from *X. exigua* are displayed in Figure 6.

Similarly, the crude extract of *X. exigua* also displayed protective activity. The cytotoxic and antioxidant activities of the crude extract of *X. exigua* collected from Pecaron Bay, Pasir Putih, Situbondo, East Java, Indonesia, were determined. The cytotoxic assay was carried out using MTT methods for cancerous cells, including HT-29, T47D, and Casky, while the antioxidant assay was conducted using the DPPH (2,2-diphenyl-1-picrylhydrazyl) method. The ethanol extract of *X. exigua* (1 mg/mL) showed cytotoxic activity with IC_50_ values of 124 (HT-29), 98 (T47D), and 68 μg/mL (Casky), while antioxidant activity was 89 μg/mL [40].

The above information displayed that the metabolites detected in *X. exigua* have a diverse range of chemical structures, including alkaloids, quinones, and sterols, with protective activity such as indoleamine-2,3-dioxygenase inhibition, vasodilation, and cytotoxicity against human cancer cell lines. Some metabolites demonstrated anti-inflammatory and antibacterial properties. Crude extract of *X. exigua* also showed cytotoxic and antioxidant properties. These results demonstrated the potential of *X. exigua* as a source of bioactive metabolites for further research.

### 2.4. Xestospongia testudinaria (Lamarck, 1815)

*X. testudinaria* is a well-known sponge species that is common across the Indo-Pacific and dominates coral reef sponge ecosystems in Indonesia. The function of the so-called “giant barrel sponge” in the reef ecology as well as its different bioactive substances have been extensively researched. Important metabolites include nonanedioic acid (azelaic acid) (**26**), which displayed bactericidal activity in a variety of Gram-positive and Gram-negative bacteria such as *Propionibacterium acnes*, *Staphylococcus epidermidis*, *S. aureus*, *Pseudomonas aeruginosa*, *Escherichia coli*, *Corynebacterium diphtheriae* and *Proteus mirabilis* [67]. It also displayed potent anti-inflammatory and antioxidant properties. Compound **26** treatment dramatically decreased the number of inflammatory papules and pustules in individuals. It promoted the transcription of genes involved in the production of proinflammatory cytokines, including IL-1β, IL-6 or TNFα [67]. Tetradecanoic acid (**27**), at concentrations of 1.0, 2.5, and 5.0 ppm, displayed larvicidal activity against *Aedes aegypti* and *Culex quinquefasciatus* mosquitoes’ larvae with LC_50_ values of 14.08 and 25.10 ppm, respectively [68]. Trans phytol (**28**), at a concentration of 10 μg/mL showed aromatase inhibition activity with an IC_50_ value of 1 μM. The compound **28** reduced aromatase mRNA and protein expression levels in human ovarian granulosa-like KGN cells [69]. In vertebrates, the sole enzyme that catalyzes the biosynthesis of estrogens is aromatase. Overexposure to estrogens causes endometrial, ovarian and breast cancer, so lowering estrogen levels by inhibiting aromatase becomes a possibility in the prevention and treatment of estrogen-mediated cancer [69,70]. In another study, **28** displayed antioxidant properties, at a concentration of 7.2 μg/mL displayed 59.89 and 62.79% scavenging capacity of DPPH• and ABTS•+ (2,2′-azino-bis(3-ethylbenzthiazoline-6-sulphonic acid), respectively. The compound **28** (administered via the intraperitoneal (i.p.) route at doses of 25, 50 and 75 mg/kg), reduced lipid peroxidation (LP) and nitrite (NO_2_^−^) levels and elevated reduced glutathione (GSH), superoxide dismutase (SOD) and catalase (CAT) activities in the Swiss mouse hippocampus [71]. Pentadecanoic acid **(29)** displayed anticancer effects against human breast carcinoma MCF-7/stem-like cells (SC). Furthermore,**29** reduced interleukin-6 (IL-6)-induced JAK2/STAT3 signalling. In populations of CD44+/CD24 stem-like cells derived from human breast cancer cells, JAK2/STAT3 signalling is essential for maintenance [72,73]. Thus, targeting JAK2/STAT3 signalling is seen as a viable therapeutic approach. Additionally, **29** caused a cell cycle arrest at the sub-G1 phase and aided caspase-dependent death in MCF-7/SC cells [72,73]. Palmitic acid (**30**) displayed anticancer properties against the human prostate cancer cell lines PC3 and DU145 as well as the subcutaneous xenograft model. Both in vitro and in vivo, **30** reduce the development of prostate cancer cells. The exposure of **30** induced G1 phase arrest, linked with the downregulation of cyclin D1 and p-Rb and upregulation of p27 [54]. The compound **30** displayed anticancer properties against murine colorectal carcinoma (CT-26) and (MC-38) cell lines [55]. It also demonstrated antiviral activity against viremia of carp virus (SVCV) infection using Zebrafish (*Danio rerio*) model. The findings showed that a low concentration of **30** modulates the infection and reduced Zebrafish mortality [56]. At concentrations ranging from 0.1 to 1.0 mg/mL, 9,12-octadecadienoic acid (linoleic acid) (31) demonstrated antibacterial activity against five Gram-positive bacteria, including *Bacillus cereus*,* B. pumilus*,* B. subtilis*,* Micrococcus kristinae*, and *S. aureus*, but was inactive against Gram-negative species (*Enterobacter cloacae*, *E. coli*, *Klebsiella pneumoniae*, *P. aeruginosa* and *Serratia marcescens*) [74]. Heneicosane (**32**) at a concentration of 10 μg/mL displayed excellent antimicrobial potential against *S. pneumoniae* (zone of inhibition = ZOI = 31 ± 0.64 mm) and *Aspergillus fumigatus* (ZOI = 29 ± 0.86 mm) respectively [75]. 1-iodohexadecane (hexadecyl iodide or cetyl iodide) (**33**) reduced 2,4-Dinitrochlorobenzene-Induced Atopic Dermatitis (AD) (a chronic inflammatory dermal) in mice. The treatment with bioactive alkane at a concentration of 100μg/mL for 21 days improved AD-like skin lesions, suppressed epidermal thickness and elevated filaggrin [76,77]. The above-mentioned long-chain fatty acids from *X. testudinaria* are displayed in Figure 7.

The above studies indicated that potential secondary metabolites were found in *X. testudinaria* with different biological activities, including anti-inflammatory, antioxidant, larvicidal, aromatase inhibition, anticancer, antiviral, and antimicrobial.

In another study, the organic (methanol—dichloromethane) extracts of *X. testudinaria* from Langkawi, Malaysia, displayed antimicrobial activity against *S. aureus*, *B. cereus*, and *E. coli* with ZOI of 11.5, 12 and 9 mm, while no antimicrobial activity was noticed in the aqueous extract of the same species against the same bacteria [45]. Similarly, the symbiotic bacteria extract isolated from *X. testudinaria* was found to possess antibacterial activity against different types of bacteria, including *S. aureus, P. aeruginosa, E. coli,* and *Salmonella typhi*, as conducted by the disc diffusion dilution method. The n-hexane, ethyl acetate, and *n*-butanol fractions of the extract demonstrated antibacterial activities. Additionally, phytochemical screening of the extract revealed the existence of important metabolites, such as alkaloids, steroids, and triterpenoids [78]. The chloroform fraction of methanol extract of *X. tesdudinaria* displayed anticancer properties against the HeLa cell line using the MTT method. The fraction displayed anticancer activity with IC_50_ values of 2.273 ppm. The GC-MS analysis indicated the presence of 21 metabolites [77]. Another study reported the antioxidant, anti-inflammatory, and immunomodulatory properties of the methanolic extract of *X. testudinaria*. against carrageenan-induced rat hind paw edema. The methanolic extract of *X. testudinaria* at a concentration of 100 mg/kg significantly decreased percentage increase in paw weight after carrageenan injection. The histopathological observation indicated that the extract administration reduced inflammatory cell infiltrate and capillary congestion. The extract boosted reduced glutathione, glutathione peroxidase, and catalase activities while decreasing malondialdehyde (MDA) and nitric oxide (NO) levels. Inflammatory cytokines such as tumour necrosis factor (TNF), interleukin-1 (IL-1), and IL-6 were also lowered [79]. The extract of *X. testudinaria* also possessed poisonous effects on brine shrimp (*Artemia salina*), with LC_50_ values ranging from 0.56 to 6.99 μM [80,81]. In addition to this, the extract of *X. testudinaria* from Bandung, Indonesia, mixed with the marine sponge *Melophlus sarasinorum* to form a scaffold for bone tissue, promoting the growth and division of bone cells. The study displayed promising results for bone tissue engineering [41].

Thus, in addition to the protective nature of the pure secondary metabolites of *X. testudinaria*, the crude extract and fractions of the sponge also displayed a wide range of protective properties, including antimicrobial, anticancer, antioxidant, anti-inflammatory, and immunomodulatory properties. Phytochemical screening of these extracts and fractions further revealed the presence of important metabolites. These results demonstrated the potential of *X. testudinaria* as a source of biologically active compounds with a range of therapeutic applications.

### 2.5. Xestospongia ashmorica (Hooper, 1984)

*X. ashmorica* is a marine sponge found in the Manila Channel off Mindoro Island in the Philippines. In 1996, researchers isolated three alkaloids from this sponge, named manzamine A (**34**), manzamine E (**35**), and manzamine F (**36**). These alkaloids were found to have cytotoxic properties against L5178 mouse lymphoma cells at a concentration range of 0.3 to 20 μg/mL. The median effective dose (ED50) for compounds **34-36** were 1.8, 6.6 and 2.3 g/mL respectively, showing that all the compounds were active against the selected cell line [42]. The structures of the compounds **34-36** from *X. ashmorica* are illustrated in Figure 8**.**

### 2.6. Xestospongia vansoesti (Bakus & Nishiyama, 2000)

The compounds salsolinol (**37**), norsalsolinol (**38**), *cis*-4-hydroxysalsolinol (**39**), and *trans*-4-hydroxysalsolinol (**40**) were isolated from the marine sponge X. vansoesti found in North Sulawesi, Indonesia. These compounds are all tetrahydroisoquinoline alkaloids. Compound (**37**) showed cytotoxicity against various cancer cell lines such as murine leukaemia (L1210) with IC_50_ values of 8 mg/mL, human amnion (FL) with IC_50_ values of 13 mg/mL, human oral epidermoid carcinoma (KB) with IC_50_ values of 20 mg/mL and human lung adenocarcinoma (A549) with IC_50_ values of 27 mg/mL, respectively. Additionally, both compounds **37** and **38** inhibited the activity of the proteasome with IC_50_ values of 50 and 32 mg/mL, respectively, and were also cytotoxic to human cancer cell lines (HeLa) with IC_50_ values of 17 and 7 mg/mL However, compounds **39** and **40** were found to have no protosome inhibitory effect and no cytotoxicity against HeLa [39]. The chemical structures of compounds **37-40** are exhibited in Figure 9**.**

## 3. Summary of *Xestospongia* Metabolites

Table 1 below summarizes the metabolites detected in the extracts of various species of *Xestospongia*, including their bioactivity, class, molecular formula (MF), and molecular weight (MW).

*Xestospongia* spp., found in Southeast Asian waters, are a rich source of secondary metabolites with potential for application in various industries, such as pharmaceuticals, biotechnology, and agriculture. Among the species, *X. exigua* displayed a high number of metabolites with bioactivity, followed by *X. testudinaria, X. bergquistia, X. ashmorica, X. vansoesti,* and *X. muta.* Despite the potential application, there is still limited knowledge about *Xestospongia* spp. and the metabolites they produce. To address this knowledge gap, Southeast Asian nations must collaborate on research. The primary objectives of the research should be the identification, classification, and biosynthetic and biotechnological potential of natural compounds from *Xestospongia* spp.

The efficiency of the bioactive secondary metabolites discovered in *Xestospongia* spp. must also be determined by clinical study. Additionally, the generation of metabolites by the host sponge and its microbial population must also be investigated individually to confirm the source of these metabolites, which will not only assist in establishing the origin of the metabolites but also shed light on the sponge’s host microbial community’s biochemistry.

Further, to aid future identification and metabolomics studies, a mass spectrometry database for *Xestospongia* spp. related metabolites should be established. This will provide a valuable resource for researchers studying these species, allowing for rapid identification and characterization of new natural products.

Overall, further research on *Xestospongia* spp. is necessary to fully understand the potential of these species and their secondary metabolites.

## 4. Conclusions

The genus *Xestospongia* within the Demospongiae class of sponges from Southeast Asian waters is a rich source of biologically active compounds. This review highlights the various structural diversity of the metabolites, including alkaloids (xestospongin, araguspongine, exiguamine, motuporamine, manzamine, salsolinol and norsalsolinol), fatty acids (nonanedioic acid, phytol, pentadecanoic acid, palmitic acid, and 9,12-octadecadienoic acid), steroids (xestobergsterol, clionasterol and 5α,8α-epidioxy-24αethylcholest-6-en-βb-ol), alkane (heneicosane and iodohexadecane), and quinones (halenaquinone and exiguaquinol), that have been isolated from *Xestospongia* spp. and found to exhibit a wide range of bioactivity. These compounds displayed antioxidant properties, which means they can protect cells from damage caused by free radicals. They also demonstrated anticancer properties, which could make them useful in the development of new cancer therapies. Additionally, the compounds displayed anti-inflammatory properties, which could be beneficial in the treatment of chronic diseases such as rheumatoid arthritis and asthma. They also indicated anti-bacterial, antiviral, antiparasitic, and cytotoxic properties.

## Figures and Tables

**Figure 1 biomolecules-13-00484-f001:**
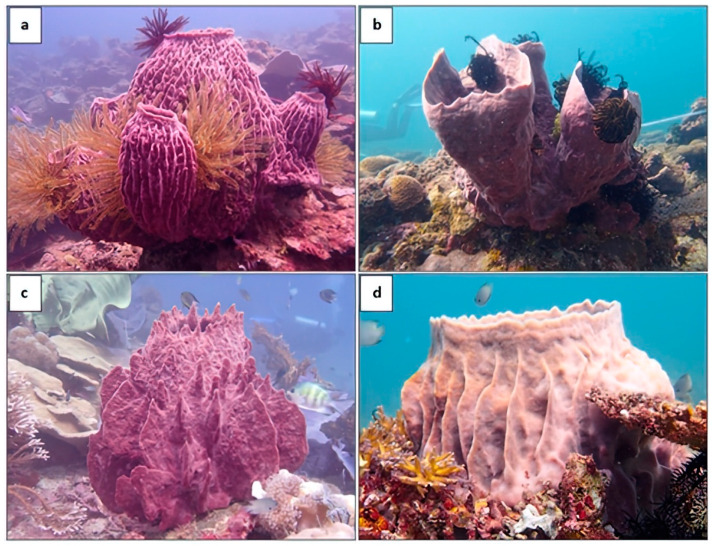
*X. testudinaria* morphotypes found in Sabah, Malaysia, waters include (**a**) digitate (outer body surface cover with digitate or spiky projections), (**b**) Smooth (lack of surface projections, (**c**,**d**) Lamellate (pronounced and smooth flanges extending from the base to the apex of its exterior).

**Figure 2 biomolecules-13-00484-f002:**
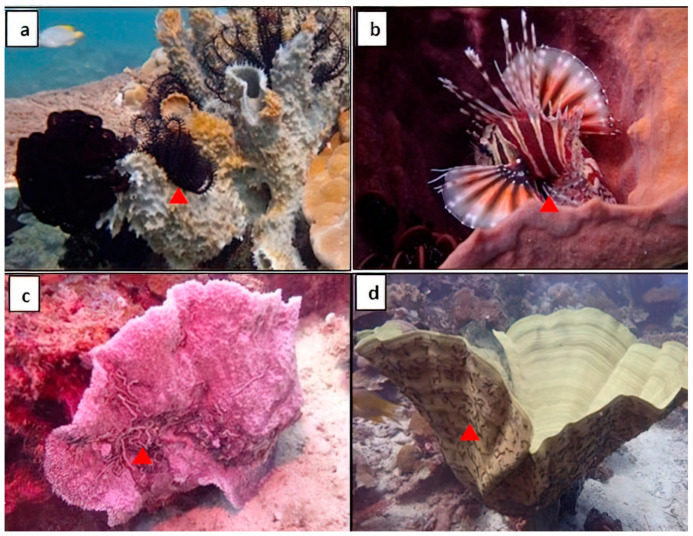
Numerous symbiotic macroscopic creatures reside inside or on the surface of a sponge. (**a**) Crinoid feather star, (**b**) Red lionfish, and (**c**,**d**) Synaptid sea cucumber.

**Figure 3 biomolecules-13-00484-f003:**
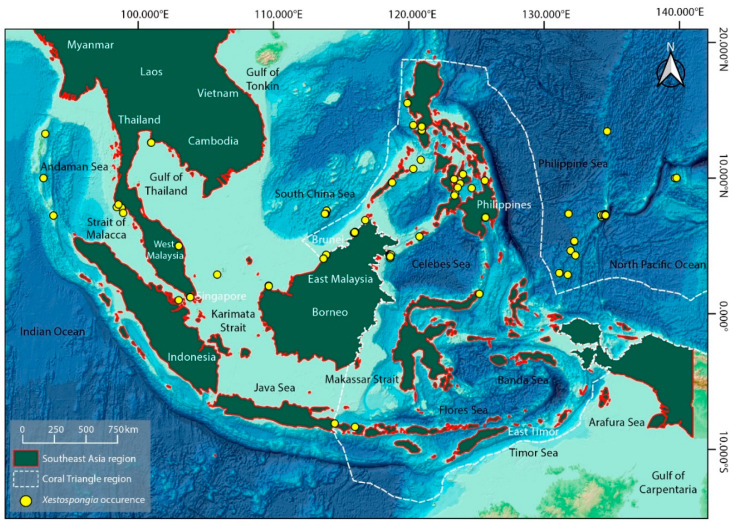
The location of *Xestospongia* spp. in Southeast Asia is indicated on the map. Malaysia, Indonesia, Thailand, Myanmar, Vietnam, Cambodia, Brunei, Singapore, Timor-Leste, and the Philippines are the nations that makeup Southeast Asia. Data for *Xestospongia* spp. (Yellow dots) are taken from the Ocean Biodiversity Information System (OBIS) (https://obis.org/; accessed on 25 December 2022) and recreated as a distribution map from the GEBCO World Map 2014. (www.gebco.net; accessed on 25 December 2022).

**Figure 4 biomolecules-13-00484-f004:**
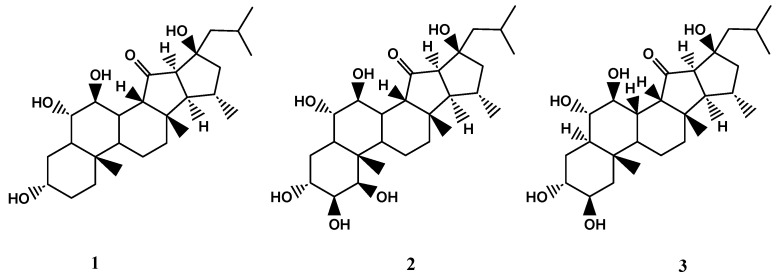
The chemical structures of xestobergsteroles A-C (**1**-**3**).

**Figure 5 biomolecules-13-00484-f005:**
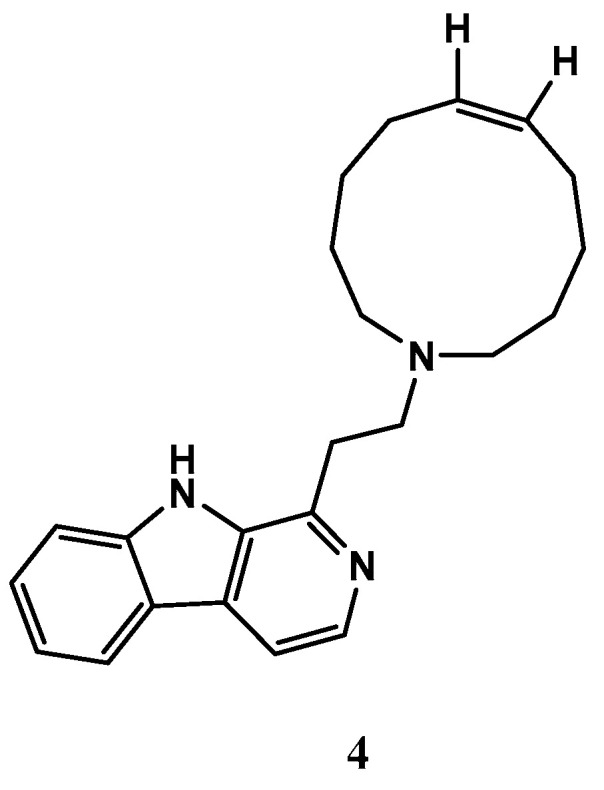
The chemical structure of manzamine C (**4**).

**Figure 6 biomolecules-13-00484-f006:**
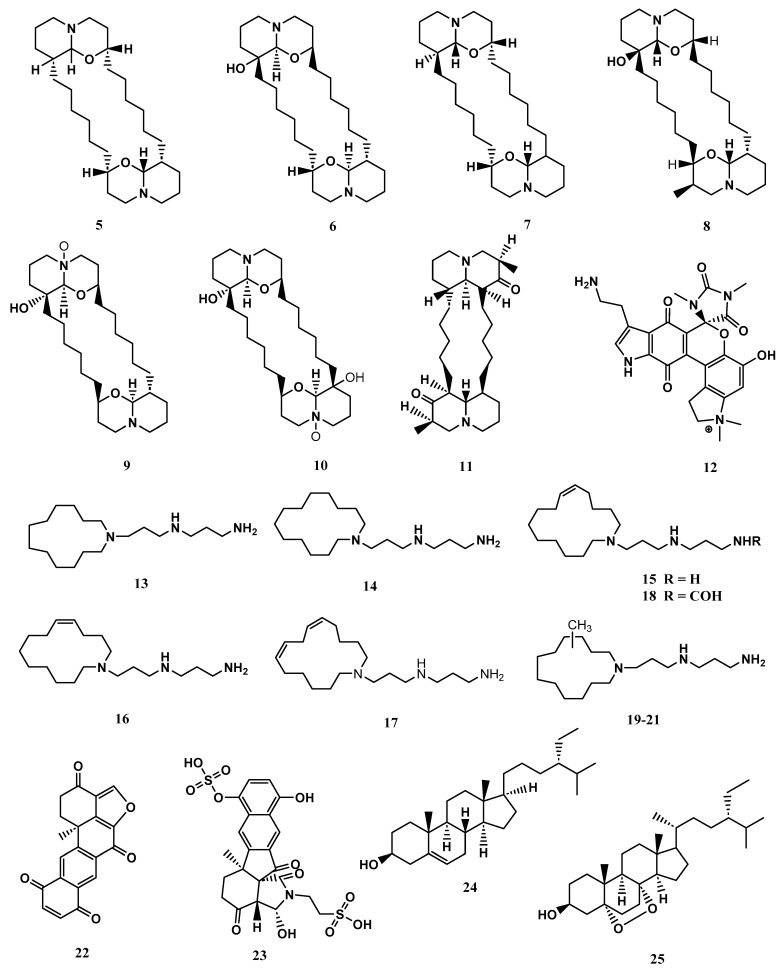
The chemical structures of compounds isolated from *Xestospongia exigua* (**5**-**25**).

**Figure 7 biomolecules-13-00484-f007:**
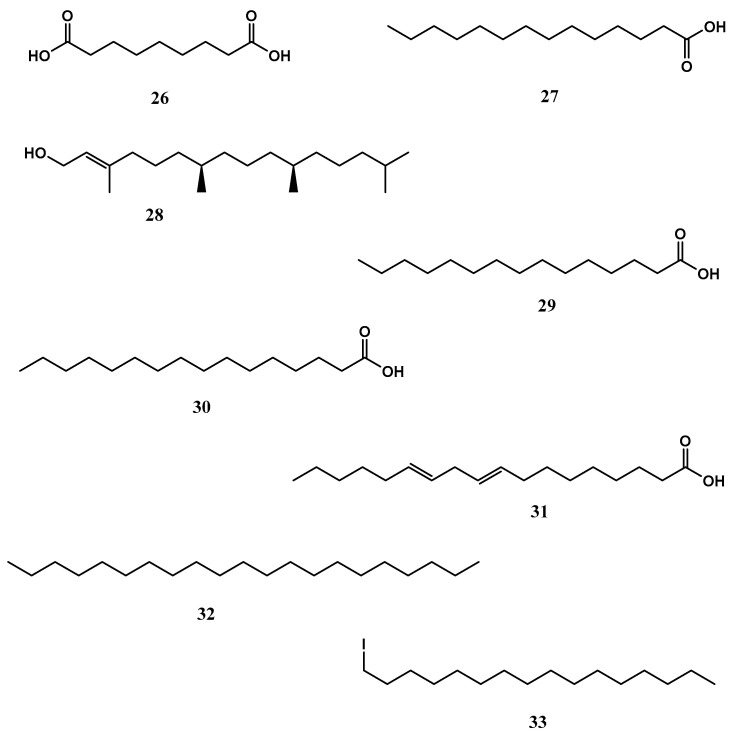
The chemical structures of long-chained fatty acids from *Xestospongia testudinaria* (**26**-**33**).

**Figure 8 biomolecules-13-00484-f008:**
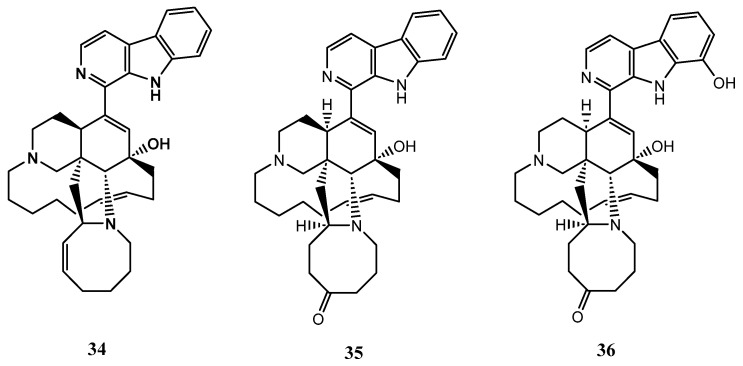
The chemical structures of manzamines A, E and F (**34**-**36**).

**Figure 9 biomolecules-13-00484-f009:**
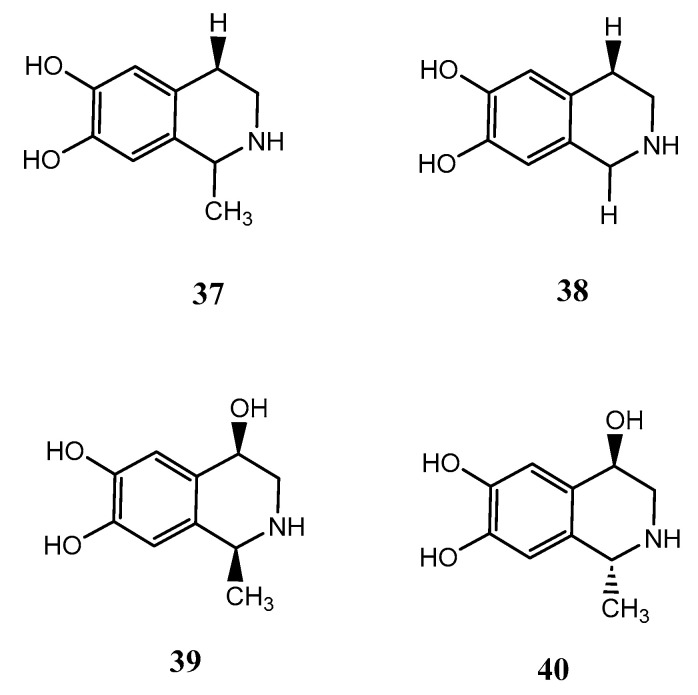
The chemical structures of salsalinol derivatives from *Xestospongia vansoesti* (**37**-**40)**.

**Table 1 biomolecules-13-00484-t001:** Summary of bioactive metabolites isolated from *Xestospongia* spp.

Species	Metabolites	Class	MF	MW	Bioactivity	References
** *X. bergquistia* **	Xestobergsterol A (**1**)	Steroid	C_27_H_44_O_5_	448.6	Anti-Inflammatory, Cytotoxic	[4,5,6,49,50]
	Xestobergsterol B (**2**)	Steroid	C_27_H_44_O_7_	480.6	Anti-Inflammatory, Cytotoxic	
	Xestobergsterol C (**3**)	Steroid	C_27_H_44_O_6_	464.6	Cytotoxic	[49,50]
** *X. muta* **	Manzamine C (**4**)	Alkaloid	C_23_H_29_N_3_	347.5	Cytotoxic	[58,59]
** *X.* *exigua* **	Xestospongin A (**5**)	Alkaloid	C_28_H_50_N_2_O_2_	446.7	Vasodilator	[62]
	Xestospongin B (**6**)	Alkaloid	C_29_H_52_N_2_O_3_	476.7	Vasodilator	
	Xestospongin C (**7**)	Alkaloid	C_28_H_50_N_2_O_2_	446.7	Vasodilator	
	Xestospongin D (**8**)	Alkaloid	C_28_H_50_N_2_O_3_	462.7	Vasodilator	
	Araguspongine K (**9**)	Alkaloid	C_28_H_52_N_2_O_4_		Vasodilation	[62,63]
	Araguspongine L (**10**)	Alkaloid	C_28_H_52_N_2_O_5_		Vasodilation	
	Exiguamine A (**12**)	Alkaloid	C_25_H_26_N_5_O_6_	492.5	Indoleamine-2,3-Dioxygenase Inhibitor	[64]
	Motuporamine A (**13**)	Alkaloid	C_18_H_39_N_3_	297.5	Cytotoxicity	[65]
	Halenaquinone (**22**)	Quinone	C_20_H_12_O_5_	332.3	Antibacterial	[52]
	Exiguaquinol (**23**)	Quinone	C_22_H_21_NO_12_S_2_	555.5	*Helicobacter Pylori* Glutamate Racemase (Muri) Inhibitor	[51]
	Clionasterol (**24**)	Steroid	C_29_H_50_O	414.7	Inhibit The Human Complement System	[53]
	5α,8α-Epidioxy-24αethylcholest-6-en-βb-ol (**25**)	Steroid	C_29_H_48_O_3_		Anti-Inflammatory, Inhibit the Human Complement System	[53,66]
** *X. testudinaria* **	Nonanedioic Acid (**26**)	Fatty acid	C_9_H_16_O_4_	188	Antibacterial And Anti-inflammatory	[67]
	TetradecanoicAcid (**27**)	Fatty acid	C_14_H_28_O_2_	228	Larvicidal	[68]
	Trans-Phytol (**28**)	Fatty acid	C_20_H_40_O	296	Aromatase Inhibitor, Antioxidant	[69,71]
	Pentadecanoic Acid (**29**)	Fatty acid	C_15_H_30_O_2_	242	Anticancer	[73]
	Palmitic Acid (**30**)	Fatty acid	C_16_H_32_O_2_	256	Antiviral, Anticancer	[54,55,56]
	9,12-Octadecadienoic Acid (**31**)	Fatty acid	C_19_H_34_O_2_	294	Antibacterial	[74]
	Heneicosane (**32**)	Alkane	C_21_H_44_	296	Antimicrobial	[75]
	1-Iodohexadecane (**33**)	Alkane	C_16_H_33_I	352	Anti-inflammatory	[76]
** *X.* ** ** *ashmorica* **	Manzamine A (**34**)	Alkaloid	C_36_H_44_N_4_O	548.8	Cytotoxic	[42]
	Manzamine E (**35**)	Alkaloid	C_36_H_44_N_4_O_2_	564.8	Cytotoxic	[42]
	Manzamine F (**36**)	Alkaloid	C_36_H_44_N_4_O_3_	580.8	Cytotoxic	[42]
** *X. vansoesti* **	Salsolinol (**37**)	Alkaloid	C_10_H_13_NO_2_	179.2	Cytotoxic	[39]
	Norsalsolinol (**38**)	Alkaloid	C_9_H_11_NO_2_	165	Cytotoxic	[39]

## Data Availability

Not applicable.

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
