# Peer review of "Metabolites and Bioactivity of the Marine Xestospongia Sponges (Porifera, Demospongiae, Haplosclerida) of Southeast Asian Waters"

_biomolecules, 2023, doi:10.3390/biom13030484_

Round 1

Reviewer 1 Report

This review collects and compares information from peer-reviewed articles from 1974 to 2022 on secondary metabolites isolated from Xestospongia spp. The authors have done a great job of searching for relevant literature, but in my opinion, the work lacks its own conclusions and generalization of the results obtained.

 Line 117. As usually the name of the compounds written with a small letter - xestobergsterol A

Line 139. ships with other organisms. [37,38]. – the point before Links

Line 134, Line 149.

This is optional, however, in my opinion, in a large review, if the figure imagine the structures of the same type or one compound, then it would be easier if the figure caption contained the names of these compounds. For example:

Figure 4. The chemical structures of xestobergsteroles A–C (13).

Figure 5. The chemical structure of manzamine C (4).

It might be a little faster to read.

Line 151. «The X. muta collected from Terengganu, Malaysia, was found to have cardiovascular protective properties»

Preferred «collected on the coast Terengganu (Malaysia)»

Line 151. Misprint (see the attachment pdf)

References (see the attachment pdf) must be checked in accordance with the journal rules.

In summary,

The review is written with the preservation of the author's style of the cited papers; accordingly, it should be assumed that these papers have passed the review stages.

However, the authors should pay attention to some wording. For example, an indication of the place of collection of the objects under study.

I also ask the authors to harmonize the references to compound numbers in the text. For example,

Line 320. The chemical structures of compounds (37 – 40).

Line 305. The chemical structures of compounds (34-36). Etc..

Another aspect, the names of connections in different places of the text with a small letter or with a capital letter.

The image of stereocenter configurations is not very nice. It would be nice if the authors could use another drawing program. The main reason, that the any bonds imagine not clear. For example, Figure 6.

In Conclusion the Authors write «The wide range of bioactivity exhibited by the compounds isolated from Xestospongia spp. suggests that they could be developed into new drugs to treat a variety of diseases». In my opinion, it is necessary to analyze the main part of the review and select the most promising molecules for their further use or study as a potential drug. Otherwise, this conclusion sounds unfounded.

Author Response

R1- Responses to the Reviewers comments

Manuscript Number: biomolecules-2231524

 â€¯ 
Title: Metabolites and Bioactivity of the Marine Xestospongia Sponges (Porifera, Demospongiae, Haplosclerida) of Southeast Asian Waters: A Mini-Review

Dear Editor,

Thank you for your valuable comments and suggestions, which are useful for improving our manuscript.

We have given our responses to all the queries raised by the reviewer.

Reviewer 1

This review collects and compares information from peer-reviewed articles from 1974 to 2022 on secondary metabolites isolated from Xestospongia spp. The authors have done a great job of searching for relevant literature, but in my opinion, the work lacks its own conclusions and generalization of the results obtained.

Q1: Line 117. As usually the name of the compounds is written with a small letter - xestobergsterol A

Res: Thank you very much for the comments. We have written the name of all the compounds with small letters except the one start after a full stop.

Q2: Line 139. ships with other organisms. [37,38]. – the point before Links.

Res: Thank you very much for the comments. Line 139 has been corrected.

Q3: This is optional, however, in my opinion, in a large review, if the figure imagine the structures of the same type or one compound, then it would be easier if the figure caption contained the names of these compounds. For example:

Figure 4. The chemical structures of xestobergsteroles A–C (13).

Figure 5. The chemical structure of manzamine C (4).

It might be a little faster to read.

Res: Thank you very much for the comment. All the figures have been renamed as below.

Figure 4. The chemical structures of xestobergsteroles A-C (1-3).

Figure 5. The chemical structure of manzamine C (4).

Figure 6. The chemical structures of compounds isolated from Xestospongia exigua (5-25).

Figure 7. The chemical structures of long-chain fatty acids from Xestospongia testudinaria (26-33).

Figure 8. The chemical structures of manzamines A, E and F (34-36).

Figure 9. The chemical structures of salsalinol derivatives from Xestospongia vansoesti (37-40).

Q4: Line 151. «The X. muta collected from Terengganu, Malaysia, was found to have cardiovascular protective properties» Preferred «collected on the coast Terengganu (Malaysia)»

Res: Thank you very much for the comments. Line 151 has been changed to “The X. muta collected in coastal Terengganu (Malaysia) was found to have cardiovascular protective properties”

Q5: Line 320. The chemical structures of compounds (37 – 40) and  Line 305. The chemical structures of compounds (34-36). Etc.

Res: Thank you very much for the comments.” Line 320. The chemical structures of compounds (37 – 40). Line 305. The chemical structures of compounds (34-36)” have been corrected.

Q6: The image of stereocenter configurations is not very nice. It would be nice if the authors could use another drawing program. The main reason, that the any bonds imagine not clear. For example, Figure 6.

Res: Thank you very much for the comments. We believe it is because the figure has too many structures. we used chem draw and improved the structures. we hope it is much better. As far as we are concerned, only Figure 6 seems to be an issue. Only this was edited.

Q7: In Conclusion, the Authors write «The wide range of bioactivity exhibited by the compounds isolated from Xestospongia spp. suggests that they could be developed into new drugs to treat a variety of diseases». In my opinion, it is necessary to analyze the main part of the review and select the most promising molecules for their further use or study as a potential drug. Otherwise, this conclusion sounds unfounded.

Res: Thank you very much for the comments.  The conclusion has been improved and the most promising metabolites have been highlighted in the conclusion.

“The genus Xestospongia within the Demospongiae class of sponges from Southeast Asian waters is a rich source of biologically active compounds. This review highlights the various structural diversity of the metabolites, including alkaloids (xestospongin, araguspongine, exiguamine, motuporamine, manzamine, salsolinol and norsalsolinol), fatty acids (nonanedioic acid, phytol, pentadecanoic acid, palmitic acid, and 9,12-octadecadienoic acid), steroids (xestobergsterol, clionasterol and 5α,8α-epidioxy-24αethylcholest-6-en-βb-ol), alkane (heneicosane and iodohexadecane), and quinones (halenaquinone and exiguaquinol), that have been isolated from Xestospongia spp. and found to exhibit a wide range of bioactivity. These compounds displayed antioxidant properties, which means they can protect cells from damage caused by free radicals. They also demonstrated anticancer properties, which could make them useful in the development of new cancer therapies. Additionally, the compounds displayed anti-inflammatory properties, which could be beneficial in the treatment of chronic diseases such as rheumatoid arthritis and asthma. They also indicated anti-bacterial, antiviral, antiparasitic, and cytotoxic properties.”

Reviewer 2 Report

The submitted manuscript entitled “Metabolites and Bioactivity of the Marine Xestospongia Sponges (Porifera, Demospongiae, Haplosclerida) of Southeast Asian Waters: A Mini-Review” summarized the metabolites identified from Xestospongia spp. distributed across Southeast Asian waters as well as their various biological activities reported. The manuscript was well drafted with some editing errors. The paper is of interest to the readers of Biomolecules and I would recommend accepting the manuscript after the comments were addressed.

In general, there is no need to capitalize the first letter of the trivial names of the reported compounds if they appear in the middle of a sentence. Please correct them throughout the whole manuscript. E.g.: L117: “Xestobergsterol” should be corrected as “xestobergsterols”. L168: “(+)-Xestospongin (A-D) (5-8)” should be corrected as “(+)-xestospongins A−D (58)”

Also, L218: “The compound (28)”, L229: “Pentadecanoic acid (29)”, L230: “Furthermore, (29)” should be corrected as “Compound 28”, “Pentadecanoic acid (29)”, and “Furthermore, 29”, respectively. Please correct them throughout the whole manuscript. The numbering of the compounds should be bold, and it should not be in a bracket if showing up alone.

Formatting of the units: g/mL and mg/mL. Please check and correct throughout the whole manuscript.

L35: Delete “indole” and “terpenes”.

L54: Correct “several” to “over 30”

 L156: “In vitro

L198: “such as”

L233: Please try to move the sentence “Additionally, …” to line 217.

L255: “are illustrated”

L270: Please double check “ The chloroform extract fraction of…”. The chloroform fraction of the crude extract? Or just the chloroform extract?

L276: “significantly decreased % increase in…” Correct % to “percentage”?

Author Response

R1- Responses to the Reviewers' comments

Manuscript Number: biomolecules-2231524

 â€¯ 
Title: Metabolites and Bioactivity of the Marine Xestospongia Sponges (Porifera, Demospongiae, Haplosclerida) of Southeast Asian Waters: A Mini-Review

Dear Editor,

Thank you for your valuable comments and suggestions, which are useful for improving our manuscript.

We have given our responses to all the queries raised by the Reviewers.

Reviewer 2

Q1: The submitted manuscript entitled “Metabolites and Bioactivity of the Marine Xestospongia Sponges (Porifera, Demospongiae, Haplosclerida) of Southeast Asian Waters: A Mini-Review” summarized the metabolites identified from Xestospongia spp. distributed across Southeast Asian waters as well as their various biological activities reported. The manuscript was well drafted with some editing errors. The paper is of interest to the readers of Biomolecules and I would recommend accepting the manuscript after the comments were addressed.

In general, there is no need to capitalize the first letter of the trivial names of the reported compounds if they appear in the middle of a sentence. Please correct them throughout the whole manuscript. E.g.: L117: “Xestobergsterol” should be corrected as “xestobergsterols”. L168: “(+)-Xestospongin (A-D) (5-8)” should be corrected as “(+)-xestospongins A−D (58)”

Also, L218: “The compound (28)”, L229: “Pentadecanoic acid (29)”, L230: “Furthermore, (29)” should be corrected as “Compound 28”, “Pentadecanoic acid (29)”, and “Furthermore, 29”, respectively. Please correct them throughout the whole manuscript. The numbering of the compounds should be bold, and it should not be in a bracket if showing up alone.

Formatting of the units: g/mL and mg/mL. Please check and correct throughout the whole manuscript.

Res: Thank you very much for the comments. We have corrected the compounds, E.g.: L117: “Xestobergsterol”  to “xestobergsterols”. L168: “(+)-Xestospongin (A-D) (5-8)” to “(+)-xestospongins A−D (58)”.

The number of the compounds has been bold and the bracket of the compounds showing up alone has been deleted.

The Format of the units has been corrected throughout the text.

Q2: L35: Delete “indole” and “terpenes”.

Res: Thank you very much for the comments. L35: “indole” and “terpenes have been deleted.

Q3: L54: Correct “several” to “over 30”

Res: Thank you very much for the comments. L54:  “several”  has been changed to  “over 30”

Q4: L156: “In vitro

Res: Thank you very much for the comments. L156: “In vitro” has been corrected

Q5: L198: “such as”

Res: Thank you very much for the comments. L198: “such as”  has been added.

Q6: L233: Please try to move the sentence “Additionally, …” to line 217.

Res:  Thank you very much for the comments. There was a mistake in sentence L233. The no of the compound was 29 not 27 and it has been corrected. The sentence is linked to compound 29, not compound 27, so it has not been changed to line 217.

Q7: L255: “are illustrated”

Res: L255: “are illustrated has been corrected.

Q8: L270: Please double check “The chloroform extract fraction of…”. The chloroform fraction of the crude extract? Or just the chloroform extract?

Res: Thank you very much for the comments. The sentence has been corrected as “ the chloroform fraction of methanol extract”

Q9: L276: “significantly decreased % increase in…” Correct % to “percentage”?

Res: Thank you very much for the comments. decreased % increase has been changed to the decreased percentage increase

Round 2

Reviewer 1 Report

The authors corrected some minor flaws and the manuscript became better.

However, in Figure 1, in structures 1-3, there is still a methyl group and a methine proton arranged so that they do not overlap each other.

Author Response

Q1: Figure 4, in structures 1-3, there is still a methyl group and a methine proton arranged so that they do not overlap each other

Res: Thank you very much for the comments. The methyl group and methine proton overlap is corrected in Figure 4, structure 1-3.